# Utilization of Plant Oils for Sustainable Polyurethane Adhesives: A Review

**DOI:** 10.3390/ma17081738

**Published:** 2024-04-10

**Authors:** Żaneta Ciastowicz, Renata Pamuła, Andrzej Białowiec

**Affiliations:** 1Department of Applied Bioeconomy, Wrocław University of Environmental and Life Sciences, 37a Chełmońskiego Str., 51-630 Wrocław, Poland; zaneta.ciastowicz@upwr.edu.pl; 2Selena Industrial Technologies Sp. z o.o., Pieszycka 3, 58-200 Dzierżoniów, Poland; renata.pamula@selena.com

**Keywords:** bio-renewable oils, environmental impact, carbon footprint, waste, polyurethane adhesives, decarbonization

## Abstract

The utilization of plant oils as a renewable resource for the production of polyurethane adhesives presents a promising way to improve sustainability and reduce environmental impact. This review explores the potential of various vegetable oils, including waste oils, in the synthesis of polyurethanes as an alternative to conventional petroleum-based raw materials. The investigation highlights the environmental challenges associated with conventional polyurethane production and highlights the benefits of switching to bio-renewable oils. By examining the feasibility and potential applications of vegetable oil-based polyurethanes, this study emphasizes the importance of further research and development in this area to realize the full potential of sustainable polyurethane adhesives. Further research and development in this area are key to overcoming the challenges and realizing the full potential of plant-oil-based polyurethanes in various industrial applications.

## 1. Introduction

Polyurethanes (PU) are one of the most important polymers because of their versatility. The primary reason is due to excellent performance properties and good adhesion for many surfaces. Because of their chosen raw materials and specific ratio, the modulus, bond strength, and speed of cross-linking can be tailored to meet the specific needs of customers [1]. Ultimately, the properties of polyurethane chains can be engineered by many possible additives and modifiers. This is a simple process and can be easily mixed into polyurethane resin. A common characteristic of these adhesives is that urethane groups are present in composition or are formed during the curing process [2].

## 2. Major Types of Polyurethane Adhesive

There are many approaches and classifications to the differentiation of adhesives, and one of them is as follows:Reactive adhesives;Solvent-based adhesives;Water-based adhesives;Hot-melt adhesives [1,3].

The reactive adhesives consist of two types in this category: one component or two components. One-component adhesives involve long molecular weight isocyanate-terminated prepolymers, which are made from polyols and diisocyanates. The urethane bonds appear as a result of the reaction between components. Using an excess of isocyanate, the characteristic feature of them is a possible reaction with ambient moisture. The free isocyanate groups react with moisture to form urea linkages. Two-component adhesives consist of low molecular weight ingredients, which are polyols and diisocyanates. The components are mixed directly before application and cured to a polyurethane. The pot life of these adhesives depends on the type of reactants and amount of catalyst [4]. Solvent-based adhesives involve high molecular weight hydroxy-terminated polymers. They are made by reaction of a polyester polyol with a diisocyanate and then the polymer is dissolved in a solvent. The curing is a physical process of evaporation of the solvent [5]. Water-based adhesives are high molecular-weight polyurethane polymers that are dispersed in water. The mechanism of curing is the same as the solvent-based systems, but the evaporated substance is water. There is a significant advantage from an environmental and human health point of view. These environmentally friendly properties make them a preferable choice, contributing to reduced emissions of volatile organic compounds (VOCs) and promoting a safer and healthier working environment for both humans and the ecosystem [6]. Hot-melt adhesives require two-step curing. The first step is the application of molten liquids and solidifying. The second step is hardening through the reaction of isocyanate groups with atmospheric moisture. Hot-melt adhesives are based on isocyanate-terminated polymers and can be formulated from high molecular weight polyester or polyether polyols with isocyanates [3,4,6].

## 3. European Adhesive Market

Adhesives play a crucial role in a huge range of consumer, professional, and industrial products, but they are mostly invisible after use. In 2020, the European adhesive market was 3.9 million tons (Figure 1), and its value was EUR 11.6 billion. The European adhesive market has been forecasted to reach EUR 16.5 billion by 2026 [7].

The demand for adhesives and sealants in the European market is expected to experience significant growth over the next five years, with the assembly market leading the expansion, followed by transportation and the building, construction, civil engineering, and craftsmen segment [7]. The Central and Eastern European region is anticipated to witness stronger growth compared to Western Europe. A notable trend involves the ongoing replacement of mechanical fasteners with chemical applications, particularly evident in the building, construction manufacturing, and assembly industries. Additionally, efforts to reduce the carbon footprint remain a priority, with a focus on energy conservation. This emphasis on sustainability positively impacts the use of adhesive bonding [8].

Adhesive bonding offers an environmentally friendly alternative to traditional mechanical fastening methods. As the global emphasis on sustainability and environmental responsibility grows, industries seek solutions that reduce their carbon footprint and minimize waste. This process eliminates the need for additional materials like screws, nuts, or bolts, which are commonly used in traditional mechanical fastening methods. Traditional mechanical fastening methods, such as welding or using fasteners, often involve energy-intensive processes. Welding, for instance, requires high temperatures and energy consumption. In contrast, adhesive bonding generally requires less energy during application. The reduced energy consumption contributes to a lower carbon footprint, aligning with global efforts to decrease greenhouse gas emissions and combat climate change [9].

In applications such as building and construction, there is a continuous trend toward using lightweight materials to enhance energy efficiency and overall performance. Adhesive bonding allows for the effective joining of lightweight materials without compromising structural integrity [8]. This contributes to fuel efficiency in transportation and resource efficiency in construction. Adhesives often serve dual purposes by providing not only structural bonding but also sealing and insulating properties. In building and construction applications, adhesives can contribute to weatherproofing, insulation, and preventing leaks, enhancing the overall performance and sustainability of structures [10].

All of the following markets are consumers of polyurethane adhesives. In packaging applications, the most important are acrylic waterborne adhesives. Ethylene-vinyl acetate (EVA) hot melts and natural polymer-based products are also broadly used. There is also space for polyurethane adhesives, which are considered one of the high-performance products offered to this industry because of their advantages, such as heat resistance, chemical resistance, and fast curing properties. In the packaging market, polyurethane adhesives are used to laminate films, foils, and paper in a variety of packaging constructions [11]. The PU systems can be used: one- and two-component water-based, solvent-free, and solvent-based adhesive, as well as specific hot melts. Polyurethane adhesives can also be designed to meet Food and Drug Administration approval, which is required for food packaging applications [6,7]. Building, construction, civil engineering, and craftsmen comprise one of the two biggest market segments in adhesive consumption. This market for polyurethane adhesives consists of a variety of applications, such as laminating wall panels, bonding gypsum board to wood ceilings, constructing montage modular and mobile houses, and gluing plywood floors. In the building market there is new work and also repair, maintenance, renovations, and flooring. The textile market is the largest consumer of polyurethane adhesives. Finally, civil engineering, with bridges, highways, railroad crossings, and pipes, most commonly uses one- and two-component systems that are 100% solid [5,7]. Woodworking and joinery are mainly cabinet making, furniture, frames for windows and doors, and upholstery. One-component and hot-melt polyurethane adhesives represent the majority of the volume in the market because of water and thermal resistance [12]. The transportation market uses polyurethane adhesives for different applications, such as bonding Fiber-Reinforced Plastic (FRP) and sheet molding composite (SMC) panels in trucks and cars. Using adhesive is also connected with polycarbonate headlamp assemblies, door panels, and weatherstrip flocking. Reactive PU adhesive is one of the main adhesive types for growth and development in the automotive sector for the next 5–10 years. Solvent-based PU adhesives are a rather niche application in automotive for rubber to substrate application but will continue to be replaced by water-based polymers [6,7]. Adhesives and sealants are used also in commercial, military, and general aviation. The consumer adhesive market includes home repair products, “do-it-yourself” (DIY) adhesives, stationery products, instant glues, wood glues, and hot-melt sticks. A significant number of adhesives come from the water-based polymers category and silane-modified polymers, acrylics, and polyvinyl acetate. The share of polyurethane adhesives is limited to wood and montage adhesives mainly because of their price [7]. Polyurethane adhesives are widely used in the footwear and leather industry due to their excellent bonding properties. Following are some common applications of polyurethane adhesives in the footwear and leather industry: sole attachment and the bonding of leather pieces together in the production of bags, belts, and wallets. A broad variety of products are sold to this market, such as solvent-based PU adhesive, polyurethane dispersions as an alternative to solvent-based systems, and reactive hot-melt PU [3,4,5]. The assembly operations market for polyurethane adhesives includes a wide range of applications across different industries, such as sandwich panels; appliances and electrical equipment; heating, ventilation, and air conditioning systems (HVAC); and alternative energy systems. One-component and two-component solvent-free adhesives and reactive hot-melt PU adhesives are some of the most commonly used products in this market [5,7].

## 4. Raw Materials for Standard Polyurethane Adhesives

Polyurethanes are formed by the reaction of a polyol with an isocyanate (Figure 2). The isocyanate groups react with the hydroxyl groups of the polyol to form a urethane linkage between the polyol molecules. The reaction is typically carried out in the presence of a catalyst, compound, and other additives may also be included to modify the properties of the resulting polymer [3].

There are two common types of commercial polyurethane adhesives: rigids and flexibles. The rigid PU products are based on polyester polyols, and flexible urethanes are based on polyether polyols [6]. Trade polyether polyols are typically synthesized through the ring-opening polymerization of epoxide monomers, like propylene oxide and ethylene oxide. Polyester polyols are generally prepared by the reaction between organic acids and alcohols. The selection of monomers for polyester polyols is more diverse compared to polyether polyols (Figure 3). This allows a broad range of properties. Examples of organic acids that can be used include adipic acid, phthalic acid, and terephthalic acid, and examples of alcohols that can be used include ethylene glycol, propylene glycol, and 1,4-butanediol [13].

Aromatic diisocyanates, such as methylene diphenyl diisocyanate (MDI) and toluene diisocyanate (TDI), are commonly used in the production of polyurethane adhesives due to their high reactivity and ability to cross-link with polyols (Figure 4) [14].

Polymeric MDI is frequently used in the production of adhesives and is a variant of MDI that consists of a mixture of MDI isomers with higher molecular weights. Polymeric MDI offers certain advantages over monomeric MDI in polyurethane adhesive applications. It typically has a higher viscosity, which can contribute to better handling and reducing migration during application [1]. The higher molecular weight of polymeric MDI can also result in improving mechanical properties and can increase resistance to chemical and thermal degradation in the cured adhesive [4]. Polymeric MDI typically has lower costs than pure MDI and has a lower freeze point (liquids at room temperature). Because of this, it has less tendency to dimerization, and as a consequence, gives a more storage stable product than pure MDI and derivatives [3]. Polymeric isocyanate type is used where the color of the finished adhesive is not important. In addition to aromatic isocyanates, aliphatic isocyanates are also used in polyurethane adhesives [5]. Aliphatic isocyanates offer certain advantages over aromatic isocyanates in certain applications. They generally have better resistance to UV light and weathering, which makes them suitable for outdoor applications where color stability and durability are important. Some commonly used aliphatic isocyanates in polyurethane adhesives include hexamethylene diisocyanate (HDI), isophorone diisocyanate (IPDI), and cyclohexane diisocyanate (H_12_MDI) [14].

Polyols and isocyanates react readily at room temperature. The reaction rates are dependent on solvent, temperature, and the presence of catalysts. Catalysts can significantly accelerate these reactions and can, in some cases, alter the order of reactivity [12]. Commercial catalysts consist of two main classes: organometallics and tertiary amines. The most commonly used catalysts in the industry are 1,4-diazabicyklo [2.2.2]oktan (DABCO), dimorpholinodiethyl ether (DMDLS), and dibutyltin dilaurate (DBTL) [5]. For many applications, adhesives based on polyols and isocyanates have too high a viscosity. The properties of adhesive can be changed by adding dry organic solvents, such as ethyl acetate, acetone, and methyl ethyl ketone [3,5]. Polyurethane adhesives can also use plasticizers to reduce viscosity, improve performances at low temperatures, and soften. Commercially, the most popular are phthalate, benzoate, phosphates, and aromatic oils. Plasticizers should be used carefully, as adhesion will generally decrease as levels of plasticizers increase [4,6].

Fillers are used in adhesives to improve physical properties, control rheology, and reduce costs. The most common polyurethane fillers are calcium carbonate, talc, silica, clay, and barium sulfate. Calcium carbonates, clays, talcs, and sulfate are used to improve the economics of an adhesive formulation. Fumed silicas are used primarily as thixotropies in application areas that require non-sagging properties [3,7]. Castor oil derivatives can indeed be used as thickeners in various applications. One common derivative used as a rheology modifier is micronized hydrogenated castor oil (also known as castor wax or castor wax flakes) [15]. The main problem with using fillers with urethane prepolymers is the moisture content associated with the fillers. Fillers usually must be dried before use with urethane prepolymers or isocyanates. Hygroscopic fillers should be avoided, as the moisture present in the filler can reduce the shelf life of the finished product [3]. Drying agents are used to avoid these problems. Common drying agents include desiccants like silica gel, calcium chloride, molecular sieves, and *p*-toluenesulfonyl isocyanate (PTSI) [5].

## 5. Environmental Issues with the Production of Polyurethane Adhesives

Production of polyurethane polymers derived from biobased feedstock has gained significant interest in recent years due to rising concerns about oil prices, global warming, and environmental issues like waste management. One of the reasons is the need for diversification of feedstock. Petrochemical-based polymers, such as those derived from crude oil, have been the dominant choice for a wide range of applications. However, relying solely on fossil fuel feedstocks poses several risks of lack of raw material. Biobased feedstocks offer an alternative by utilizing renewable resources such as plant-based biomass, agricultural waste, and plant oils. This diversification reduces the dependence on limited fossil fuel reserves and promotes sustainability and mitigation of carbon footprint [16,17,18]. Renewable substrates for polyurethane production can significantly lower the environmental impact associated with traditional petrochemical-based polymers. By utilizing renewable feedstocks, these polymers can reduce greenhouse gas emissions and contribute to mitigating climate change [19].

Life Cycle Assessment (LCA) is commonly used to evaluate the effects of biobased materials in comparison to their petrochemical equivalents to deliver a numerical response and validate environmental assertions. According to the most recent EN ISO Standards 14040:2009 and EN ISO 14044:2009 series, LCA serves as a standardized method for gauging the potential environmental implications linked with a product or service. This process includes gathering relevant inputs and outputs, evaluating the potential environmental consequences associated with these inputs and outputs, and interpreting the results of both the inventory and impact phases of this study’s goals [20,21].

There are several well-established methods for life-cycle impact assessment developed in European centers, which relate data to average conditions in Europe. These include EPS 2000, CML, Eco-indicator 99, IMPACT 2002+, ReCiPe, and MIPS. These methods are implemented in computer programs used for LCA techniques, such as SimaPro, GaBi, and Umberto [22]. Life Cycle Assessment (LCA) analysis demonstrates that utilizing rapeseed oil as a biobased raw material for polyol production results in significant reductions in environmental impact compared to using petrochemical polyols. These benefits include lower non-renewable energy consumption, reduced greenhouse gas emissions, and decreased water usage [23]. Life cycle assessments show clear environmental benefits for flexible foam polyols made primarily from soy or castor oil compared to petrochemicals. The seed oil-based polyols use fewer fossil resources and could generate very low greenhouse gas emissions. These oils have a reduced regional air impact, emit lower levels of sulfur oxides (SOx) and nitrogen oxides (NOx) than the European average, and do not require chlorine-based chemistry, reducing the need for infrastructure and the potential creation of unwanted chlorinated organic by-products [24].

Nonpetrochemical polymers can contribute to the development of a circular economy by enabling the production of materials that can be recycled, composted, or safely returned to nature after use [25,26].

### 5.1. European Union Green Deal-General Issue

In December 2019, the European Council launched the European Green Deal. It is a package of policy initiatives that aim to set the EU on the path to a green transition, with the ultimate goal of reaching climate neutrality by 2050. Climate change has been established by European Climate Law. The EU aims to reduce greenhouse gas emissions by at least 55% by 2030 and achieve climate neutrality by 2050. It involves increasing the share of renewable energy, improving energy efficiency, and implementing measures to reduce emissions from industry, transport, and buildings. The EU aims to promote a circular economy model, which involves minimizing waste, reusing and recycling materials, and reducing resource consumption. The goal is the transition from a linear “take-make-dispose” approach to a more sustainable and circular production and consumption system [7,27].

The EU Chemicals Strategy for Sustainability is a comprehensive plan developed by the European Commission to ensure the safe and sustainable use of chemicals in the European Union. Its main objective is to protect human health and the environment while promoting the competitiveness of the EU chemicals industry. Key goals and principles of the EU Chemicals Strategy for Sustainability include safer chemicals, better protection of human health, enhancement of the industry competitiveness, and support of a toxic-free environment [7,27,28,29]. Trends such as the following have emerged to meet EU requirements: decarbonization and reduction of the carbon footprint. They refer to the global shift away from carbon-intensive activities and the reduction of greenhouse gas emissions, particularly carbon dioxide (CO_2_), to mitigate climate change and transition toward a more sustainable and low-carbon economy. The trends involve various strategies and actions aimed at reducing carbon emissions across different sectors, such as energy, transportation, industry, and agriculture [27]. The principal aspects of the trends include renewable energy transition, implementation of energy efficiency measures, sustainable transportation, sustainable agriculture and food systems, circular economy, and waste reduction [28,29].

### 5.2. Decarbonization and the EU Green Deal in the Construction Market

The building construction and operation sector accounts for a significant portion of energy consumption and greenhouse gas emissions in Europe. Therefore, the European Green Deal recognizes the need to make buildings and all sectors more energy-efficient, sustainable, and environmentally friendly [30]. Here are some key aspects and initiatives related to the European Green Deal in building construction:The EU aims to accelerate the renovation of existing buildings to improve their energy efficiency. The Renovation Wave initiative seeks to double the renovation rate of buildings across Europe and prioritize energy-efficient upgrades, including insulation, efficient heating and cooling systems, and the use of renewable energy sources [31].The Energy Performance of Buildings Directive (EPBD) sets standards for energy performance in buildings and promotes the use of renewable energy sources [32].Sustainable Building Standards: The European Green Deal emphasizes the development and promotion of sustainable building standards and certifications through: Leadership in Energy and Environmental Design (LEED) and BREEAM are widely recognized and applied across Europe to ensure that buildings meet high environmental performance standards [32].The Green Deal promotes the principles of the circular economy in the construction sector. This involves minimizing waste generation, promoting the reuse and recycling of materials, and adopting more sustainable construction practices [28].The EU is mobilizing significant financial resources to support the transition to greener buildings through the European Investment Bank, and various funding programs aim to provide financial incentives and support for energy-efficient building renovations and sustainable construction projects [7].The European Green Deal finally promotes research and innovation in the construction sector to develop new materials, technologies, and construction methods that are more sustainable and energy-efficient [27].

The EU strategy of the Green Deal has a more direct impact on the European Adhesive Market. The main adhesive sustainable trends are the demand for solvent-free adhesives, the development of compostable adhesives, and biobased adhesives where petrochemical raw materials will be replaced by raw materials of plant origin or from waste [7,27].

## 6. Plant Oils Resources for Eco-Friendly Polyurethanes

Various vegetable oils are renewable sources used during the manufacturing of raw materials for polyurethane adhesives, and they can be successfully and extensively used as a substitute petrochemical material [33].

Many directions for making plant oil based on polyols are described, including chemical reactions like epoxidation, transesterification, ozonolysis, thiol-ene coupling reaction, hydroformylation, and photochemical oxidation. It is worth mentioning that the same types of chemical reactions can be used for processing waste vegetable oils and waste from the agricultural and food industries. The most common possibilities of reaction are seen in Table 1 [26,34,35].

A review of the literature revealed that there are a lot of publications on the synthesis of biobased polyurethanes. Almost all consist of biobased polyols or biobased plasticizers and unfortunately, in almost all of them, petrobased isocyanates were used [51,52,53,54,55,56,57,58,59,60,61,62]. There are commercially available isocyanates Desmodur Eco N7300 (70% renewable carbon content) and Desmodur CQ 44V20 L (60% renewable carbon content) from Covestro (Leverkusen, Germany) [63,64]. The literature available concerning the synthesis of isocyanate compounds based on plant oils is limited. There is research on creating isocyanates from soybean oil and canola oil [65,66]. The synthesis of suitable diisocyanates derived from vegetable oils is still a challenge. Because of the toxicity of isocyanate, the more popular is to obtain polyurethanes by using the nonisocyanate route [67,68,69].

Also, biobased plasticizers can be obtained mainly from plant oil, agricultural products, and waste. The main conversion paths include the following chemical reactions: transesterification [70,71], hydrogenation [72,73], and epoxidation [37,74]. Some plant oils can be added directly to polymers in definite percentages to enhance their properties [75].

### 6.1. The Use of Plant Oil

The following Table 2 focuses on the main available vegetable oils and waste materials that potentially can be used as biobased raw materials for polyurethane adhesives.

The comprehensive Table 2 puts forward the use of various vegetable oils and waste materials as potential biobased raw materials for polyurethane production. Including application details and investigating polyurethane properties offers a complete overview of the research conducted in this field. Testing properties, such as cure characteristics, thermal properties, mechanical properties, density, and various spectroscopic analyses like FTIR and NMR, demonstrates a thorough assessment of these raw materials’ suitability for polyurethane production [51,52,56,57,59,61]. A thorough perspective is crucial for understanding the potential advantages and limitations of incorporating renewable natural resources into sustainable polyurethanes. Several studies have highlighted the enhanced properties of polyurethanes when utilizing biobased materials [52,53,54,55,75]. Orgilés-Calpena et al. emphasized the superior properties of soy-based PU compared to petroleum-based counterparts. Products offer enhanced flexibility and elasticity and additionally, the composition of soy-based polyols can be adjusted to customize the properties of the PU, allowing tailored solutions to meet specific application requirements [52]. Fridrihsone-Girone et al. outlined the beneficial properties of rapeseed oil polyols in their study. The polyurethane coatings exhibited enhanced hydrolytic stability and minimal water absorption and also demonstrated commendable physical and mechanical properties, meeting the requirements for two-component polyurethane coatings according to ASTM standards [53]. Członka et al. mentioned the use of linseed oil as a natural modifier in rigid polyurethane foams, which can lead to improvements in mechanical strength, foam morphology, thermal properties, and reactive properties, potentially affecting fire resistance properties [54]. Jatropha oil, derived from the non-edible Jatropha curcas plant, has been reported to yield products with improved biodegradability compared to traditional petroleum-based polyols. Moreover, Jatropha oil can be chemically modified to improve compatibility with other polymers and additives, thus enhancing the overall performance and properties of the resulting polyurethane foam [55]. Conversely, deterioration was observed in the properties in the following publications: [53,56,58,76]. Khoon Poh et al. observed that the properties of palm-oil-based polyols may have some challenges in terms of thermal stability and mechanical performance compared to traditional wood adhesives [56]. In another study, one property that was found to be worse with rapeseed oil polyols was the hydrolytic stability of coatings containing polyol formulation [53]. Kirpluks et al. identified some properties that may be worse with oil polyols, including lower mechanical, dimensional, and thermal stability compared to petrochemical-based polyols. High-functioning polyols are required for rigid PU foam production, and biobased polyols may have limitations in achieving the required functionality for certain applications [76]. Another study found that sunflower oil polyols have inferior properties compared to conventional polyols, including lower tensile strength, lower elongation at break, lower impact resistance, and slightly lower scratch hardness [58]. It is important to note that the quantity of vegetable oil or waste material used significantly influences the outcomes of the research [53,58]. Moreover, the table underscores the increasing focus on decarbonization and the use of renewable resources in the polymer industry. In conclusion, the table acts as a valuable resource for comprehending the diverse range of plant oils and waste materials under investigation, along with the specific tests being evaluated in the context of polyurethane composition production.

### 6.2. Potential of Waste Oil Used for Polyurethane Production

The food industry, gastronomy, and households collectively purchase 400,000 tons per year of oil from producers. Out of this total, 90% is fully utilized, leaving 10%, equivalent to 40,000 tons per year, as waste. It is estimated that only around 10,000 tons per year of waste fats find their way to specialized recipients for proper disposal [77]. Used oils from the agricultural and food industry may present an ecological problem especially if not properly disposed of and processed. Following are some issues that can impact the environmental aspect of waste oils:Water Pollution and Issues with Sewer Systems. Improper disposal of used oils, such as dumping in sewers or wastewater, can lead to water pollution. This poses a threat to aquatic ecosystems by affecting water quality and the survival of aquatic organisms. Additionally, if used oils are flushed into the sewer, they can contribute to the formation of fatbergs, leading to blockages in sewage pipes and system failures, incurring additional repair costs [78,79,80].Waste of Resources. If used oils are not collected and processed for reuse, they represent a waste of potential resources. Proper processing can contribute to a closed material loop and reduce waste [78,81,82].

To mitigate the negative impact of used oils on the environment, promoting proper waste management practices, recycling, and exploring alternative uses for these oils as a resource for raw materials are crucial. Raising ecological awareness and implementing appropriate regulations can help minimize the adverse effects of used oils on the environment.

The use of plant oils and waste for the production of biobased polyurethanes has many advantages, but unfortunately, it is not without disadvantages. Following are some of the drawbacks associated with the use of biobased oils:Biobased oils are typically derived from crops. Their production is limited by the availability of suitable land, water resources, and favorable climatic conditions. Consequently, it might compete with food production, leading to concerns about food security and increased prices for agricultural commodities [83,84].The cultivation of crops for biobased oils can result in increased pressure on land resources. Deforestation not only reduces biodiversity but also releases significant amounts of carbon dioxide into the atmosphere, aggravating climate change [84].Biobased oil production requires water for irrigation, and the increased demand for water can strain local water supplies. This can lead to conflicts over water usage among agricultural needs, local communities, and ecosystems [23,85].While biobased oils are often considered more environmentally friendly than fossil fuels, their production and processing can still generate greenhouse gas emissions. The cultivation, harvesting, and transportation of crops, as well as the extraction and processing of oils, can contribute to carbon dioxide emissions and other greenhouse gases [84,86].Intensive cultivation of biobased oil crops can lead to soil erosion, nutrient depletion, and reduced soil quality. Without proper management practices, the continuous cultivation of oilseed crops can degrade soil fertility and contribute to land degradation [87,88].

In conclusion, the study of oil utilization and waste management in the food industry reveals critical environmental challenges due to inadequate proper disposal. Improper handling poses ecological threats like water pollution, highlighting the need for effective waste management. Addressing this issue requires increased awareness and strict regulations for proper disposal. Recognizing waste oils as valuable resources is crucial. Proper processing contributes to promoting sustainable development and implementing the principles of circular economy. While there are advantages to using vegetable oils for biobased polyurethanes, careful consideration is needed to balance the benefits with potential impacts on food security, deforestation, water scarcity, and emissions. Future research should focus on innovative technologies to minimize the environmental impact of waste oils. Finding alternative sources of biobased oils that do not compete with food production is essential for a sustainable future.

At this point, it is important to mention the very important environmental challenge of used motor oil [89]. Waste oil can be re-refined into lubricants, processed into fuel oils, and used as raw materials for the refining and petrochemical industries [90]. It is also possible to valorize the vacuum residue of motor oil by adding it to bitumen [91]. Polyurethane-modified bitumen waterproofing membranes are well known [92].

## 7. Summary and Future Trends

In summary, the integration of plant oils and waste oil from the agricultural industry into polyurethane chemistry offers significant financial and ecological benefits, contributing to both environmental protection and production sustainability.

The availability of renewable natural resources, unlike petrochemical resources, is virtually limitless, leading to a surge in research publications focusing on the development of biobased polyols. While much of the existing research focuses on conventional reactions for new biobased polyols, cost-effective solutions remain a priority. It is crucial to recognize that the production and processing of biobased oils can contribute to greenhouse gas emissions across their life cycle, stemming from various stages like land-use changes, agricultural practices, and energy requirements. An innovative approach is to use vegetable oils as natural modifiers in building products without chemical modification, which promises to reduce the carbon footprint. This strategy, which uses vegetable oils in their least processed form, can maximize environmental benefits. Rigorous testing of mixtures with varying weight contents of additives is essential to understand their effects on application and mechanical properties. The review emphasizes the potential of vegetable oils, including waste cooking oil, to enhance sustainability. Research into oils as natural modifiers is highlighted as a means of further improving the environmental profile of polyurethane adhesives. This aligns with the broader goals of promoting circular economy principles and reducing waste in polyurethane production.

However, a critical assessment of the environmental impact is paramount. While plant oils offer a renewable alternative, their cultivation, processing, and utilization may still pose environmental challenges. Addressing potential trade-offs associated with plant oil-based polyurethanes is essential for informed decision-making. Investigating oils as natural modifiers offers new opportunities to improve the environmental performance of polyurethane adhesives. Integrating life cycle assessment (LCA) studies into evaluations can provide a comprehensive understanding of their environmental performance throughout their life cycle. By exploring waste cooking oil and plant oils as natural modifiers, this study contributes to developing environmentally friendly materials and processes, aligning with broader sustainability goals.

This research is motivated by the urgent need to address the environmental challenges of conventional polyurethane production and to promote sustainable practices. By exploring waste cooking oil and plant oils as natural modifiers, this study contributes to developing environmentally friendly materials and processes, aligning with broader sustainability goals.

## Figures and Tables

**Figure 1 materials-17-01738-f001:**
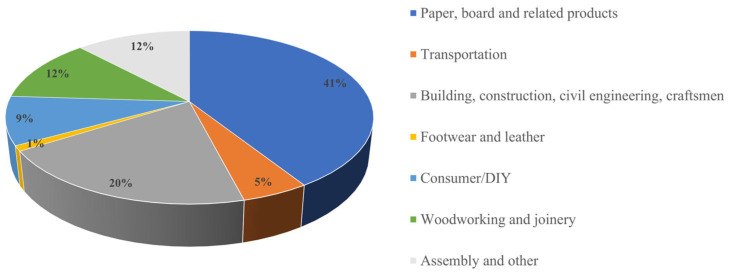
European adhesive consumption in 2020.

**Figure 2 materials-17-01738-f002:**
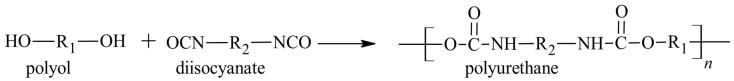
Reaction between polyol and diisocyanate.

**Figure 3 materials-17-01738-f003:**
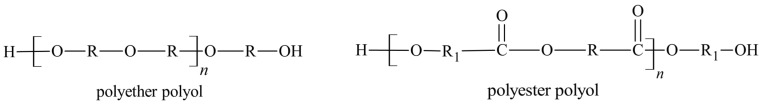
Structure of polyols.

**Figure 4 materials-17-01738-f004:**
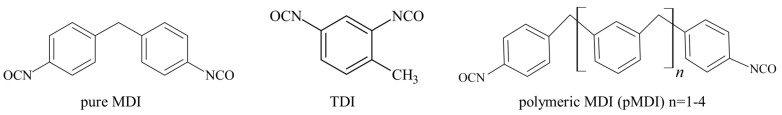
Structure of isocyanates.

**Table 1 materials-17-01738-t001:** Synthesis method of plant-oil-based polyols.

Route Modification	Number of Production Steps	State of Art	References
epoxidation	2	commercial	[36,37,38,39,40]
transesterification	1	commercial	[41,42,43,44,45]
ozonolysis	2	investigation	[46]
thiol-ene coupling	1	investigation	[47]
hydroformylation,	2	investigation	[48,49]
photochemical oxidation	2	investigation	[50]

**Table 2 materials-17-01738-t002:** Vegetable Oils and Waste Materials as Potential Biobased Raw Materials.

Plant Oil	Application	Investigated Polyurethane Properties	References
Castor oil	polyol/adhesive	cure characteristic, TGA, shore A hardness, density, tensile properties, DSC analysis,	[51]
Soybean oil	polyol/adhesive	DSC analysis, TGA, FTIR analysis, GPC analysis, mechanical properties	[52]
Rapeseed oil	polyol/coating	cure characteristic, rheology properties, density, tensile properties, FTIR analysis, hydrolytic stability tests, shore A hardness	[53]
Linseed oil	plasticizer/foam	FTIR analysis, DSC, TGA, morphological and optical properties, mechanical properties, density, dimensional stability, contact angle, water absorption, flammability	[54]
Jatropha oil	polyol/foam	FTIR analysis, TGA/DTG, SEM analysis, mechanical properties, content of closed–cell, average cell diameter, density, biodegradability	[55]
Palm oil,	polyol/adhesive	cure characteristic, chemical resistance, green strength, FTIR analysis, TGA, tensile properties	[56]
Coconut oil	polyol/foam	rheology properties, NMR analysis, FTIR analysis, GPC analysis, morphological and optical properties, DSC, TGA, thermal conductivity, cure characteristic, mechanical properties	[57]
Sunflower oil,	polyol/coating	NMR analysis, FTIR analysis, TGA, surface morphology, tensile properties, chemical resistance, biodegradability	[58]
Corn oil	polyol/foam	cure characteristic, density, and content of closed–cell mechanical properties, optical properties SEM, fire-retardant properties, TGA	[59]
Tall Oil	polyol/foam	density, content of closed–cell, thermal conductivity, mechanical properties, average cell size,	[76]
Rubber seed oil	polyol/adhesive	SEC analysis, FTIR analysis, NMR analysis, tensile properties, DSC analysis, TGA	[49]
Waste Cooking Oils	polyol/foam	foaming properties, morphology properties SEM, apparent density, thermal conductivity coefficients, compressive strength, TGA, dimensional stability	[61]
Olive oil	polyol/foam	FTIR analysis, NMR analysis, SEC analysis, rheology properties, DSC analysis, TGA, compression tests, the cell sizes, apparent foam density	[62]

## Data Availability

Not applicable.

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
