# Peer review of "Utilization of Plant Oils for Sustainable Polyurethane Adhesives: A Review"

_materials, 2024, doi:10.3390/ma17081738_

Round 1

Reviewer 1 Report

Comments and Suggestions for Authors

In this manuscript, the authors reviewed the application prospects of various vegetable oils, including waste oils, as an alternative to traditional petroleum-based feedstocks in synthetic polyurethanes. In addition, the author also reviewed the adhesive market in Europe and the environmental problems posed by adhesives, and introduces the materials that are mainly used as polyurethane based raw materials, demonstrating the possibility of using vegetable oils and waste.

1. What are the advantages and disadvantages of vegetable oil-based polyurethanes, and what are the specific performance differences compared to traditional polyurethanes, if possible, please add.

2. More information on the use of used motor oil could be added to explain the possibility of its reuse.

3. Life cycle assessment is mentioned in the manuscript, but it is not discussed in detail.

Author Response

The answers to Rewiers's comments are in the attached file.

Reviewer 2 Report

Comments and Suggestions for Authors

The authors have presented a nice review towards the use of PU with plant oils as major compounds. However, I think they are dedicating too much of the manuscript to the PUs in general and not in the plant oils as is the key point of this review. Finally, more references are needed.

Author Response

(The authors gave the same response as above.)
